# Fostering proactive work behavior: Where to start?

**A. Yuspahruddin[1], Hafid Abbas[1], Indra Pahala[1], Anis Eliyana [2]*, Zaleha Yazid [3]**

**1** Postgraduate School, Universitas Negeri Jakarta, East Jakarta, DKI Jakarta, Indonesia, **2** Department of Management, Universitas Airlangga, Surabaya, East Java, Indonesia, **3** Faculty of Economics and Management, School of Management, Universiti Kebangsaan Malaysia, Bangi, Selangor, Malaysia

☯ These authors contributed equally to this work.
* anis.eliyana@feb.unair.ac.id

**Data Availability Statement:** All files are available from the Mendeley database: Dataset: https://data.mendeley.com/datasets/gy6yx952

**Funding:** The author(s) received no specific funding for this work.

## Abstract

This study underscores the significance of assessing the capabilities of rehabilitation officers in navigating challenges, devising innovative work methods, and successfully executing the rehabilitation process. This is particularly crucial amid the dual challenges of overcapacity and the repercussions of the Covid-19 pandemic, making it an essential area for research. To be specific, it aims to obtain empirical evidence about the influence of proactive personality and supportive supervision on proactive work behavior, as well as the mediating role of Role Breadth Self-efficacy and Change Orientation. This research was conducted on all rehabilitation officers at the Narcotics Penitentiary in Sumatra, totaling 272 respondents. This study employs a quantitative method via a questionnaire using a purposive sampling technique. The data was subsequently examined using the Lisrel 8.70 software and Structural Equation Modeling (SEM). It can be concluded from the results that the rehabilitation officers for narcotics addicts at the Narcotics Penitentiary can create and improve proactive work behavior properly through the influence of proactive personality, supportive supervision, role breadth self-efficacy, and change orientation. The study may suggest new ways of working and generate new ideas to increase initiative, encourage feedback, and voice employee concerns. Furthermore, this research has the potential to pinpoint deficiencies in proactive work behavior, serving as a foundation for designing interventions or training programs. These initiatives aim to enhance the innovative and creative contributions of rehabilitation officers in the rehabilitation process.

## Introduction

The cases of drug usage have opened the eyes of all nations. The United Nations Office on Drugs and Crime (UNODC) estimates that at least 271 million individuals, or 5.5% of the world's population, between the ages of 15 and 64 have used drugs [1]. In the meantime, the National Narcotics Agency in Indonesia has stated that the nation's drug problem demands continual attention and heightened monitoring from all sectors.

**Competing interests:** The authors have declared that no competing interests exist.

The Directorate General of Corrections conducts medical and social rehabilitation for penitentiarys/detention residents. However, the penitentiarys have experienced overcapacity by 79% since the number of prisoners throughout Indonesia is 243,572 while the total capacity is only 135,075 people. Of this number, the number of bookies, custodians, dealers, producers, and narcotics users is 128,390 people or around 52.7% of the total number of prisoners.

Many prisons stopped rehabilitation activities due to entry access restrictions in response to the COVID-19 pandemic. Unfortunately, this condition did not make officers try to find solutions or ideas on how to keep the rehabilitation process going, for example, by changing the way of work and the methods of counseling, nor to convey these obstacles to their superiors. Thus, it could be concluded that the rehabilitation officers did not work proactively, even though it had been regulated in the Treatment for Prisoners SMR (Standard Minimum Rules) articles 46 and 47.

Establishing proactive work behavior requires an appropriate approach involving several other factors. A proactive personality tends to increase the ability to find new ways of doing work, including new ideas to improve initiative and function [2]. Ideally, an officer has the inner certainty to succeed at each step; it is important to foster role breadth self-efficacy because it relates to how individuals feel they can play a broader and more proactive role [3]. If this is not convincing, supportive supervision in each decision-making process will certainly give better results. Supportive supervision is a condition that reflects the extent to which team supervisors display supportive behavior [4] by encouraging feedback and voicing concerns among employees [5]. Furthermore, to display innovative behavior by creating the right ideas, officers need to focus on change orientation that will lead to making changes related to policies, methodologies, and work procedures that also occur in roles outside of their work authorization and responsibilities while bringing benefits for the organization [6]. It is intended so that officers can pass the required innovation process through proactive work behavior.

Pre-research results about proactive work behavior were carried out on Heads of correctional divisions throughout Indonesia, and 64% of them stated that rehabilitation officers did not have a proactive work behavior. As many as 30 respondents also said that the rehabilitation officers could not yet provide new innovative ideas. Meanwhile, 31 respondents stated that rehabilitation officers did not have the creativity to solve problems during this pandemic.

We are in the industrial era of 4.0, where innovation is a must. Apart from the importance of innovation in the technical aspect, focusing on individuals is certainly much more important because individuals play a bigger role in making a difference and creating innovative output among business actors [7]. About 80% of new or improvised ideas in the company are the initiative of employees, and the rest are the result of innovation plans set by the company [8].

From the above problems, it can be seen that the rehabilitation target will be difficult to achieve because the implementers are not doing enough innovation and creation, so the rehabilitation process does not run properly. It can be assumed that the rehabilitation implementers have not taken sufficient initiative to develop new work methods, have not taken the initiative to solve the problems at hand, and have not been able to take over to control the problematic conditions. Therefore, from the descriptions above, the researcher considers Proactive Work Behavior to be a gap among rehabilitation officers.

Because of its interconnectedness, this research has a sense of urgency that sets it apart from previous studies. It provides a global perspective by connecting the worldwide problem of drug usage with the local setting in Indonesia and illuminating the nation's unique problems. Moreover, the focus on the overcapacity problem in penitentiarys adds a crucial dimension, as it poses systemic challenges requiring innovative solutions, directly impacting the execution of rehabilitation programs. By exploring the role of rehabilitation officers as innovators in the industry 4.0 era, this research provides a contemporary perspective that connects

the necessity for innovation with technological advancements and shifts in work methodologies. Additionally, the study emphasizes the significance of internal motivation, spontaneity, and creativity in carrying out rehabilitation tasks, highlighting proactive work behavior as a key research variable.

This research is not only important for addressing problems within penitentiarys, but also has significant consequences for rehabilitation results and community safety. Through the integration of these aspects, the research provides a thorough and all-encompassing contribution to the comprehension of rehabilitation methods in penitentiarys. It provides comprehensive analysis of both local and global factors, enabling advancements in rehabilitation procedures and continuous refinements. This highlights crucial factors that can foster creativity in a period characterized by ongoing progress.

This research explores the crucial role of rehabilitation officers as innovators in the context of the Industrial Era 4.0, which is marked by a strong focus on innovation. It aims to clarify how these officers may effectively contribute to driving positive change. The pressing nature of this investigation creates a foundation for formulating legislation, designing training strategies, and implementing interventions aimed at improving the efficiency of rehabilitation personnel. The primary objective of this research is to enhance rehabilitation results and enhance community safety by implementing focused activities and making well-informed decisions.

The objective of this study is to investigate the proactive work behavior of rehabilitation officer in penitentiarys, with a particular focus on their initiative and innovative capabilities. Furthermore, it aims to evaluate the proficiency of rehabilitation officer in overcoming hurdles, devising new strategies, and efficiently carrying out rehabilitation procedures, especially when confronted with difficulties such as overcapacity and the COVID-19 epidemic. The study aims to investigate the elements that influence and promote proactive work behavior in rehabilitation officers, including proactive personality, supportive supervision, role breadth self-efficacy, and change orientation.

## Literature review

### Proactive work behavior

Proactive work behavior is an anticipatory action that includes seizing opportunities, preventing problems, and making independent efforts to improve and change the work context or situation and one's future [9]. Individuals with proactive work behavior act before future events, usually do not need to be asked to do something and can require less detailed instructions to achieve an action. Proactive work behavior may also indicate an individual's inclination to modify and reject things at work [10]. Proactive work behavior is self-initiated and future-oriented action to alter and enhance one's present position or oneself [11,12].

### Proactive personality

A proactive personality is a steady tendency to take the initiative in diverse settings and activities [13]. Proactive employees are inclined to offer new task performance methods and produce fresh ideas to enhance their initiative and performance [2]. Employees with proactive personalities are valuable in the workplace because they will use effort, perseverance, and effort to form a good environment [14]. These employees will improve the situation by behaving as more than passive recipients [15].

## Supportive supervision

Supervision involves the process of directing and supporting officer to be able to carry out their duties [16] effectively. Supportive supervision is a condition that reflects the extent to which team supervisors display supportive behavior [4]. It will be able to encourage feedback and voice the concerns of its employees [5]. Supportive supervision is a homogeneous structure due to the integration of regular contacts and concepts within the team [4]. It is not only because team members are subject to the same structural impact but also because supportive supervision concepts stand out from general experience.

## Role breadth self-efficacy

Role breadth self-efficacy refers to the degree to which individuals believe they can execute a larger and more proactive role [3]. It is an extension of self-efficacy related to one's belief in the potential to create and develop ideas to achieve goals [17]. Role breadth self-efficacy has a broader focus than other forms of self-efficacy, usually only related to certain tasks or activities [18].

## Change orientation

Change orientation of employees can be associated with creating changes related to policies, methodologies, and work procedures that are either included or outside their work authority while bringing constructive benefits to the organization [6]. Regarding the required innovation process, change-oriented employees can display challenging, innovative behaviors [12]. Change-oriented behavior can involve positive recognition of unexpected new environments or lead to changes that contribute to organizational effectiveness [19].

## Hypothesis development

**Proactive personality and proactive work behavior.** From a selection perspective, one of the options that organizations have is to look for employees who inherently have a proactive personality. A proactive personality is an important antecedent for proactive work behavior [20]. Individuals with a proactive personality tend to change their environment in a positive way and try to improve/enhance behavior at work [21]. Referring to the model of behavioral concordance, a highly proactive personality encourages the formation of proactive behavior [22]. The model of behavioral concordance also provides an explanation of the increased sense of competition among individuals with highly proactive personalities. In addition, these individuals have a tendency to self-start and change their own work routines because their self-concordances are higher than those with low proactive personalities [23]. Through a proactive personality, employees can introduce new procedures, suggest alternative ways to increase work effectiveness, provide visible evidence to support their belief, and have the power to control the situation needed, which in turn leads to proactive work behavior [24]. Being involved in proactive work behavior indicates a person's efforts to influence their environment and can strengthen their sense of competence when the behavior produces constructive changes [24]. In addition, a previous study has stated that a proactive personality directly affects proactive work behavior [25]. Prior research has also demonstrated the association between proactive personality and proactive work behavior in the domains of control, voice, individual creativity, and issue prevention [26].

*H1*: *Proactive Personality has a significant effect on Proactive Work Behavior*

**Supportive supervision and proactive work behavior.**   Supportive supervision is renowned for physically and psychologically assisting subordinates in dealing with their job and expectations [27]. Employees can accept this support as extra-role behavior that gives them new hope in the workplace, which is done above and beyond the description of the supervisor's formal job. Furthermore, employees can become cognitively addicted to work problems and behaviorally spend substantial personal time on their jobs due to a higher intention to engage in proactive role behaviors that lead to proactive work behavior. Although empirical research investigating the role of supportive supervision on proactive work behavior is still limited [11], there are other studies that have proven the contribution of supervision or leadership aspects in encouraging proactive behavior. According to previous research, servant supervision is able to encourage and facilitate individuals to behave proactively [28]. In addition, inclusive leadership is also known to correlate with proactive work behavior [29]. Theoretically, Social Exchange Theory (SET) is the basis for building a supportive supervision role for proactive work behavior. Meanwhile, SET emphasizes reciprocity due to interactions between two parties [30], which in this context are supervisors and their subordinates. In this case, supervision provides support and appreciation for the work of subordinates, which is rewarded by subordinates with proactive behavior. The good treatment received from the vision in the form of support will also be responded to well by followers [31–33], namely proactive work behavior. The interaction between supportive supervision and subordinates encourages individual and team discussions so that they can predict problems and generate collective ideas. Prior study also stated that when innovation supervision is highly supportive, team members are more likely to begin and continue in creative behavior and to coordinate their innovative efforts with others, which may be achieved through proactive work behavior [4]. It is in line with a previous study, which states that the support of leaders in terms of (encouragement, availability of leaders, and non-interference) positively supports employee work behavior [34].

*H2*: *Supportive Supervision has a significant effect on Proactive Work Behavior*

**Role breadth self-efficacy and proactive work behavior.**   Role breadth self-efficacy will give their employees the confidence to face proactive consequences, and some empirical studies support the notion that role breadth self-efficacy is a predictor of proactive work behavior [35]. Moreover, role breadth self-efficacy has been deemed significant since it promotes initiative and self-direction, which is advantageous for proactive work behavior [3]. Previous studies have proven that role breadth self-efficacy can inspire individuals to believe that they can carry out broader and more proactive roles [36]. It goes beyond predetermined technical criteria and produces better proactive work behavior. It puts forward two greatest concern characteristics: the ability to apply active thinking and proactively solve problems at work. An individual with role breadth of self-efficacy tends to focus on various proactive work behaviors and integrative and interpersonal relational activities that form an extended role, such as dealing with long-term problems [18]. In line with previous studies, it has also been confirmed that role breadth of self-efficacy is a significant antecedent or predictor of proactive work behavior [18,35].

*H3*: *Role Breadth Self-efficacy has a significant effect on Proactive Work Behavior*

**Change orientation and proactive work behavior.**   Employees can occasionally reinterpret organizational goals to develop more demanding objectives and actively influence the socialization process to enhance the quality of their work experience, indicating that they have

a change orientation [37]. Furthermore, they will try to display 'active' behavior by showing proactive work behavior. Employees can carry out a series of change orientation activities by bringing about changes, including changes in a situation (for example, introducing new work methods and influencing organizational strategies) and changes in themselves (for example, learning new skills to deal with future demands) where it leads to proactive work behavior [37]. Furthermore, change orientation is positively related to proactive work behavior. A change in atmosphere or targets in the organizational environment makes employees more proactive in work behavior.

*H4: Change Orientation has a significant effect on Proactive Work Behavior*

**Proactive personality and role breadth self-efficacy.** A proactive personality can affect the role breadth of self-efficacy in individual adjustment through self-efficacy components related to representative experience, performance achievement, verbal persuasion, and emotional arousal [38]. Role breadth self-efficacy assesses employees' ability to carry out various integrative and interpersonal tasks well [35]. A proactive personality can take action, play a more flexible role, and make changes similar to mastery or control [3]. It can be a person's ability to foster role breadth of self-efficacy by making people believe they can play a broader role. Previous research has shown that proactive personality is well connected to role breadth self-efficacy in a substantial way [36]. It is consistent with the findings of [39], which indicated that a proactive personality is favorably associated with self-efficacy role breadth. According to the findings, proactive personality and creativity have a beneficial influence on creative behavior and, consequently, organizational effectiveness.

*H5: Proactive Personality has a significant effect on Role Breadth Self-Efficacy*

**Supportive supervision and role breadth self-efficacy.** Supportive supervision can be an important environmental factor affecting creativity [40], which can also further influence employee self-efficacy because self-efficacy describes individual behavior as a result of the interaction between individual traits and their work environment [41]. Supportive supervision will show concern for the feelings and needs of its employees by not only encouraging them to express their views but also giving them positive information feedback, which can facilitate their abilities [40]. That way, these employees will be the strongest determinant of their tendency to suggest good ideas through role breadth self-efficacy [42]. A previous study conducted at hospital agencies stated that supportive supervision is directly related to role breadth self-efficacy [43].

*H6: Supportive Supervision has a significant effect on Role Breadth Self-Efficacy*

**Proactive personality and change orientation.** A highly proactive personality will help individuals actively seek new ideas and take the initiative to improve their situation. They are more likely to change their situation by way of an individual rather than let themselves be shaped by their environment [44]. It will lead to a state of change orientation because behavior with change orientation in organizations provides a positive job perception of individual initiatives that can also encourage adaptive behavior consistent with the organization's goals and values [19]. According to a previous study, change orientation, as influenced by a proactive personality, will also lead to innovation or action directed to implement changes in improving work practices, usually driven by personal initiative [45]. A proactive personality is directly related to change orientation [46]. This personality will develop an environment of invention

and adaptability, which can provide settings that inspire proactivity regardless of employee differences.

*H7*: *Proactive Personality has a significant effect on Change Orientation*

**Supportive supervision and change orientation.**   When supervisors practice supportive supervision, they will encourage feedback and the expression of concerns and demonstrate care for employees' needs and feelings to offer positive feedback, promote creativity, and assist in skills development [5]. That way, individuals who receive supportive supervision will feel they have the power to step up to achieve better organizational goals. They will also lead to a change orientation, which is very important in encouraging organizational change for a sustainable organization [19]. According to a previous study, supportive supervision is directly related to change orientation [47]. With the supervision of supportive supervisors, the researchers' analysis is highly correlated with the officers in the organization.

*H8*: *Supportive Supervision has a significant effect on Change Orientation*

**Mediating role of role breadth self-efficacy.**   The association between proactive personality and creative behavior is entirely mediated by employee development in the workplace, and this study demonstrates the existence of high-involvement human resource practices to boost individual proclivities and cultivate proactive work behavior [48]. Role breadth of self-efficacy can refer to a more proactive ability to do a task [49]. In addition, role breadth of self-efficacy is a state that can change with the organizational environment and employee experiences, so it is hoped that it can be carried out simultaneously with a proactive personality to form proactive work behavior. That way, the company will achieve its goals to control and change the organization's internal environment [35].

Leaders who support employees facing work-family conflicts can contribute to increasing the resources employees need to resolve work stress due to work-family conflicts [50]. Employees with role breadth self-efficacy can perform a certain series of tasks and feel motivated to do them, which, if done with employees who have supportive supervision, will be more efficient in leading to proactive work behavior. Supportive supervision can give individuals useful feedback, advice, practical assistance, and resources [4]. Hence, employees will have motivation through role breadth self-efficacy and good support through supportive supervision in influencing proactive work behavior.

*H9*: *Role Breadth Self-Efficacy significantly mediates the effect of Proactive Personality on Proactive Work Behavior*

*H10*: *Role Breadth Self-Efficacy significantly mediates the effect of Supportive Supervision on Proactive Work Behavior*

**Mediating role of change orientation.**   Change orientation is related to proactive work behavior [35]. A change in atmosphere or targets in the right organizational environment will make employees more proactive. It will increase proactive work behavior. Through behavior with change orientation, it will take a proactive personality toward problems and be involved in changes that make the tendency to be assertive and dominant [51]. That way, employees with proactive personalities will be equipped with the right change process to create proactive work behavior.

Previous studies show that the support of secure base leaders significantly positively affects proactive work behavior [34]. Leader support (encouragement, leader availability, and non-interference) positively supports employee work behavior. Furthermore, changes in the environment in an organization require the support of superiors so that proactive work behavior can continue. According to a previous study, today's work experience cannot secure a position for the future, so behavior with sustainable change orientation is very important for employees [34]. Behavior with change orientation will refer to how employees initiate changes, and it can also be supported by supportive supervision to form better proactive work behavior. This support will be an innovation for team members in starting and surviving the changes needed to create proactive work behavior.

*H11*: *Change Orientation significantly mediates the effect of Proactive Personality on Proactive Work Behavior*

*H12*: *Change Orientation significantly mediates the effect of Supportive Supervision on Proactive Work Behavior*

The following Fig 1 summarizes the hypotheses development.

## Research methods

### Research approach

This research methodology is quantitative, in which statistical data analysis was performed. Exogenous variables in this study include proactive personality and supportive supervision. in this study, the endogenous variable is proactive work behavior, while the mediating variables are role breadth self-efficacy and change orientation.

### Measurement

In this study, the independent variables are proactive personality (X1) and supportive supervision (X2), and the dependent variable is proactive work behavior (Y). Role breadth self-efficacy (Z1) and change orientation (Z2) are mediating variables. The measuring scale for respondents' responses is the five-point Likert scale, which ranges from 1 (strongly disagree) to 5 (strongly agree). The proactive personality (X1) items were derived from a previous study [52]. Furthermore, the items for the supportive supervision (X2), role breadth self-efficacy (Z1), change orientation (Z2), and proactive work behavior (Y) were sourced from a previous study [46].

### Data collection techniques

The questionnaires for this study used the Google Forms platform, which were sent to pre-selected research participants. The participants in this study were all of the 272 rehabilitation officers at the Narcotics Penitentiary in Sumatra. Furthermore, the sample size in this study was determined using a purposive sampling technique that involved obtaining samples based on specific criteria. A sample of rehabilitation officers from the Indonesian Penitentiary served as the study's criterion.

As this is non-interventional research, the Research Ethics Committee at Universitas Airlangga has determined that no ethical approval is necessary. Furthermore, following the organization's policy, the organization's Chief Executive Officer has provided written informed consent representing all the participants. Before completing the questionnaires, all respondents were also informed that data collection was not mandatory and agreed upon the written

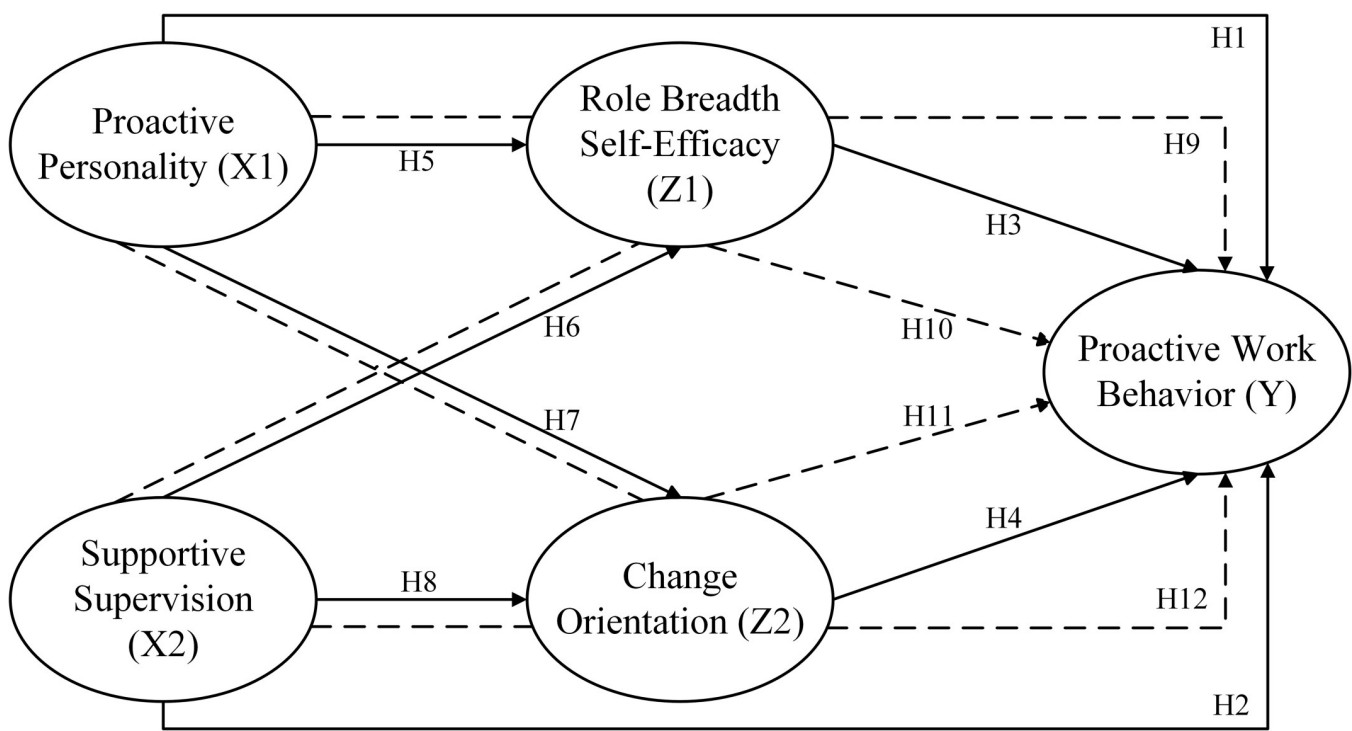

**Fig 1. Conceptual framework here.**

consent. In addition, respondents were informed that their responses were guaranteed to be kept strictly confidential and used only for research purposes.

**Table 1. Description of respondent characteristics.**

| Characteristics of Respondents | Note | Frequency | Percentage |
|---|---|---|---|
| Gender | Male | 254 | 93.4 |
| | Female | 18 | 6.6 |
| Age | 20–30 y.o | 219 | 80.5 |
| | 31–40 y.o | 46 | 16.9 |
| | 41–50 y.o | 6 | 2.2 |
| | >50 y.o | 1 | 0.4 |
| Tenure | 1–2 years | 128 | 47.1 |
| | 3–4 years | 73 | 26.8 |
| | 5–6 years | 9 | 3.3 |
| | 7–8 years | 13 | 4.8 |
| | 9–10 years | 13 | 4.8 |
| | 11–12 years | 12 | 4.4 |
| | 13–14 years | 7 | 2.6 |
| | > 8 years | 3 | 1.1 |
| | >15 years | 14 | 5.1 |
| Higher Education | Senior high | 191 | 70.2 |
| | Diploma | 17 | 6.3 |
| | Bachelor | 62 | 22.8 |
| | Master | 2 | 0.7 |

**Table 2. Multivariate normality test results.**

| Testing | Kurtosis | c.r *multivariate* | Conclusion |
|---|---|---|---|
| *Multivariate normality* | 472.532 | 62.637 | c.r. are outside the range ± 1.96, so the multivariate data are not normally distributed |

## Data analysis techniques

The data analysis technique in this study used Structural Equation Modeling (SEM) with the Lisrel 8.70 program to test and analyze the hypotheses that had been formulated.

## Results and discussion

### Results

Variable assessments were carried out by all rehabilitation officers at the Sumatra Narcotics Penitentiary by answering various statements contained in distributed questionnaires.

Table 1 shows a description of the characteristics of the rehabilitation officers for narcotics addicts in the Narcotics Penitentiary in this study; most of them are men (93.4%), aged 20–30 years (80, 5%), have been working for 1–2 years (47.1%), and the higher education degree is senior high school (70.2%).

Table 2 shows the results of the c.r multivariate normality test, which indicates that multivariate data are not normally distributed. The c.r value of 62.64 is outside the range of -1.96 to +1.96 at a significance level of 5%.

Table 3 shows the Z-score value for each indicator, showing that the Z-Score falls inside the range of 3. Hence, it is univariate. The conclusion is that none of the study's findings were recognized as outliers.

Table 4 shows the results of multivariate outlier detection; 17 observations have a d-squared Mahalanobis value> 77.42 table chi-square limit, so the 17 observations (respondents) can be indicated as outliers and then drop from the analysis. Thus, the number of samples will be reduced from 17 to 255. This sample size still meets the SEM criteria because it is still in the 215–430 range according to the requirements previously described.

Table 5, which explains Figs 2 and 3, shows that the most common absolute index is the GFI, and the most common incremental index is the CFI, which is insensitive to the influence of model complexity. Then, it is known that the evaluation of the fit measurement model produces unacceptable criteria, namely GFI, AGFI, and PNFI, which are still included in the poor fit criteria. Thus, the measurement model was further revised by modifying the model based on the value of modification indices. The examination of the measurement model's (revised model) suitability yielded criteria that were all acceptable (excellent fit and marginal fit) and more suitable than those of the initial model, making the measurement model acceptable.

Table 6 demonstrates that in the revised model, all constructs had loading factor values> 0.50 and AVE values> 0.50; therefore, they may be utilized to develop models.

Table 7 demonstrates that each variable yields a construct reliability value> 0.70. Therefore, all indicators were reliable.

After the measurement model analysis, the structural model analysis begins with a fit evaluation to ensure the constructed model matches the data (fit). Fig 4 shows the final model's estimated results and goodness-of-fit value.

Based on Table 8, the model is acceptable since all the requirements for absolute fit indices, incremental fit indices, and parsimony fit indices have been met (marginal fit and good fit). The next step is to test the direct and indirect interactions between variables.

**Table 3. Univariate outlier test results.**

| Variable | Indicator | Z-score | | Terms | Information |
|---|---|---|---|---|---|
| | | Min. | Max. | | |
| *Proactive Personality* | PP1 | -2.847 | 0,689 | -3 < Z-score < 3 | *nonoutlier* |
| | PP2 | -1.459 | 1,118 | -3 < Z-score < 3 | *nonoutlier* |
| | PP3 | -2.864 | 0,765 | -3 < Z-score < 3 | *nonoutlier* |
| | PP4 | -2.509 | 0,756 | -3 < Z-score < 3 | *nonoutlier* |
| | PP5 | -2.276 | 0,781 | -3 < Z-score < 3 | *nonoutlier* |
| | PP6 | -2.754 | 1,083 | -3 < Z-score < 3 | *nonoutlier* |
| | PP7 | -2.710 | 1,077 | -3 < Z-score < 3 | *nonoutlier* |
| | PP8 | -2.478 | 0,755 | -3 < Z-score < 3 | *nonoutlier* |
| | PP9 | -2.462 | 0,961 | -3 < Z-score < 3 | *nonoutlier* |
| | PP10 | -2.471 | 1,091 | -3 < Z-score < 3 | *nonoutlier* |
| *Supportive Supervision* | SP1 | -1930 | 0,818 | -3 < Z-score < 3 | *nonoutlier* |
| | SP2 | -2.518 | 0,698 | -3 < Z-score < 3 | *nonoutlier* |
| | SP3 | -2.016 | 0,775 | -3 < Z-score < 3 | *nonoutlier* |
| | SP4 | -2.143 | 0,854 | -3 < Z-score < 3 | *nonoutlier* |
| | SP5 | -1.971 | 0,785 | -3 < Z-score < 3 | *nonoutlier* |
| | SP6 | -2.120 | 0,771 | -3 < Z-score < 3 | *nonoutlier* |
| | SP7 | -2.040 | 0,820 | -3 < Z-score < 3 | *nonoutlier* |
| | SP8 | -2.320 | 0,714 | -3 < Z-score < 3 | *nonoutlier* |
| | SP9 | -2.006 | 0,814 | -3 < Z-score < 3 | *nonoutlier* |
| | SP10 | -1.989 | 0,785 | -3 < Z-score < 3 | *nonoutlier* |
| *Role Breadth Self-Efficacy* | RBSE1 | -2.038 | 0,919 | -3 < Z-score < 3 | *nonoutlier* |
| | RBSE2 | -2.956 | 0,992 | -3 < Z-score < 3 | *nonoutlier* |
| | RBSE3 | -2.554 | 1,058 | -3 < Z-score < 3 | *nonoutlier* |
| | RBSE4 | -2.486 | 1,000 | -3 < Z-score < 3 | *nonoutlier* |
| | RBSE5 | -2.919 | 1,005 | -3 < Z-score < 3 | *nonoutlier* |
| | RBSE6 | -2.233 | 1,068 | -3 < Z-score < 3 | *nonoutlier* |
| | RBSE7 | -2.849 | 1,166 | -3 < Z-score < 3 | *nonoutlier* |
| *Change Orientation* | CO1 | -2,631 | 0,921 | -3 < Z-score < 3 | *nonoutlier* |
| | CO2 | -2,105 | 1,244 | -3 < Z-score < 3 | *nonoutlier* |
| | CO3 | -2,558 | 0,693 | -3 < Z-score < 3 | *nonoutlier* |
| | CO4 | -2,073 | 1,244 | -3 < Z-score < 3 | *nonoutlier* |
| | CO5 | -2,276 | 1,157 | -3 < Z-score < 3 | *nonoutlier* |
| *Proactive Work Behavior* | PWB1 | -2,189 | 0,744 | -3 < Z-score < 3 | *nonoutlier* |
| | PWB2 | -1,704 | 0,998 | -3 < Z-score < 3 | *nonoutlier* |
| | PWB3 | -2,528 | 1,189 | -3 < Z-score < 3 | *nonoutlier* |
| | PWB4 | -2,132 | 0,782 | -3 < Z-score < 3 | *nonoutlier* |
| | PWB5 | -2,763 | 1,118 | -3 < Z-score < 3 | *nonoutlier* |
| | PWB6 | -2,823 | 1,095 | -3 < Z-score < 3 | *nonoutlier* |
| | PWB7 | -2,442 | 0,662 | -3 < Z-score < 3 | *nonoutlier* |
| | PWB8 | -1,782 | 0,911 | -3 < Z-score < 3 | *nonoutlier* |
| | PWB9 | -1,833 | 0,899 | -3 < Z-score < 3 | *nonoutlier* |
| | PWB10 | -1,744 | 0,899 | -3 < Z-score < 3 | *nonoutlier* |
| | PWB11 | -1,813 | 0,942 | -3 < Z-score < 3 | *nonoutlier* |

**Table 4. Multivariate outlier test results.**

| Observation number | Mahalanobis d-squared | p1 | p2 |
|---|---|---|---|
| 61 | 113,059 | ,000 | ,000 |
| 272 | 106,965 | ,000 | ,000 |
| 141 | 106,470 | ,000 | ,000 |
| 255 | 99,971 | ,000 | ,000 |
| 202 | 94,954 | ,000 | ,000 |
| 182 | 94,756 | ,000 | ,000 |
| 215 | 93,300 | ,000 | ,000 |
| 46 | 90,029 | ,000 | ,000 |
| 172 | 88,383 | ,000 | ,000 |
| 120 | 87,369 | ,000 | ,000 |
| 199 | 86,743 | ,000 | ,000 |
| 97 | 86,132 | ,000 | ,000 |
| 226 | 85,584 | ,000 | ,000 |
| 222 | 83,097 | ,000 | ,000 |
| 170 | 81,965 | ,000 | ,000 |
| 40 | 81,516 | ,000 | ,000 |
| 153 | 77,604 | ,001 | ,000 |
| 75 | 77,137 | ,001 | ,000 |
| 213 | 77,082 | ,001 | ,000 |
| 246 | 76,822 | ,001 | ,000 |
| 267 | 76,733 | ,001 | ,000 |
| 204 | 76,560 | ,001 | ,000 |
| 35 | 76,087 | ,001 | ,000 |
| 212 | 75,907 | ,001 | ,000 |
| 91 | 74,783 | ,002 | ,000 |
| 209 | 73,988 | ,002 | ,000 |
| 228 | 73,805 | ,002 | ,000 |
| 98 | 73,600 | ,003 | ,000 |
| : : | | | |
| 240 | 52,503 | ,152 | ,000 |
| 96 | 52,030 | ,163 | ,000 |

Table 9 demonstrates the impact of provisions on variables. If the CR value is 1.96 or the p-value is 5% real level, there is a significant influence between these variables. There is a negligible influence if the CR value is 1.96 or the p-value is > 5% real level.

Table 10 demonstrates the impact of provisions on variables. If the CR value is 1.96 or the p-value is 5 percent actual level, there is a substantial influence; otherwise, there is an inconsequential effect.

## Discussion

The initial hypothesis is confirmed (H1 accepted). It is consistent with findings from previous studies that a proactive personality directly affects proactive work behavior [25]. The findings of this study suggested that rehabilitation officers at the Narcotics Penitentiary could offer new methods and provide alternatives to boost their work effectively if they have a proactive personality. In addition, rehabilitation officers could offer proof to support their conviction that they were influential and could influence the required circumstance, which led to proactive

**Table 5. Fit measure on the measurement model.**

| Fit Measure | | Critical Value | Initial Model | | Revised Model | |
|---|---|---|---|---|---|---|
| | | | Index value | Decision | Index value | Decision |
| Absolute Fit Indices | Probability | ≤ 0,05 | 0,000 | Good fit | 0,000 | Good fit |
| | Cmin/DF | ≤ 2,00 | 1,959 | Good fit | 1,460 | Good fit |
| | GFI | ≥ 0,90 | 0,769 | Poor fit | 0,827 | Marginal fit |
| | RMSEA | ≤ 0,08 | 0,060 | Good fit | 0,043 | Good fit |
| | SRMR | ≤ 0,05 | 0,049 | Good fit | 0,045 | Good fit |
| Incremental Fit Indices | CFI | ≥ 0,95 | 0,907 | Marginal fit | 0,958 | Good fit |
| | TLI | ≥ 0,95 | 0,901 | Marginal fit | 0,954 | Good fit |
| | NFI | ≥ 0,90 | 0,827 | Marginal fit | 0,879 | Marginal fit |
| | RFI | ≥ 0,90 | 0,817 | Marginal fit | 0,868 | Marginal fit |
| Parsimony Fit Indices | AGFI | ≥ 0,90 | 0,743 | Poor fit | 0,801 | Marginal fit |
| | PNFI | ≥ 0,90 | 0,779 | Poor fit | 0,803 | Marginal fit |

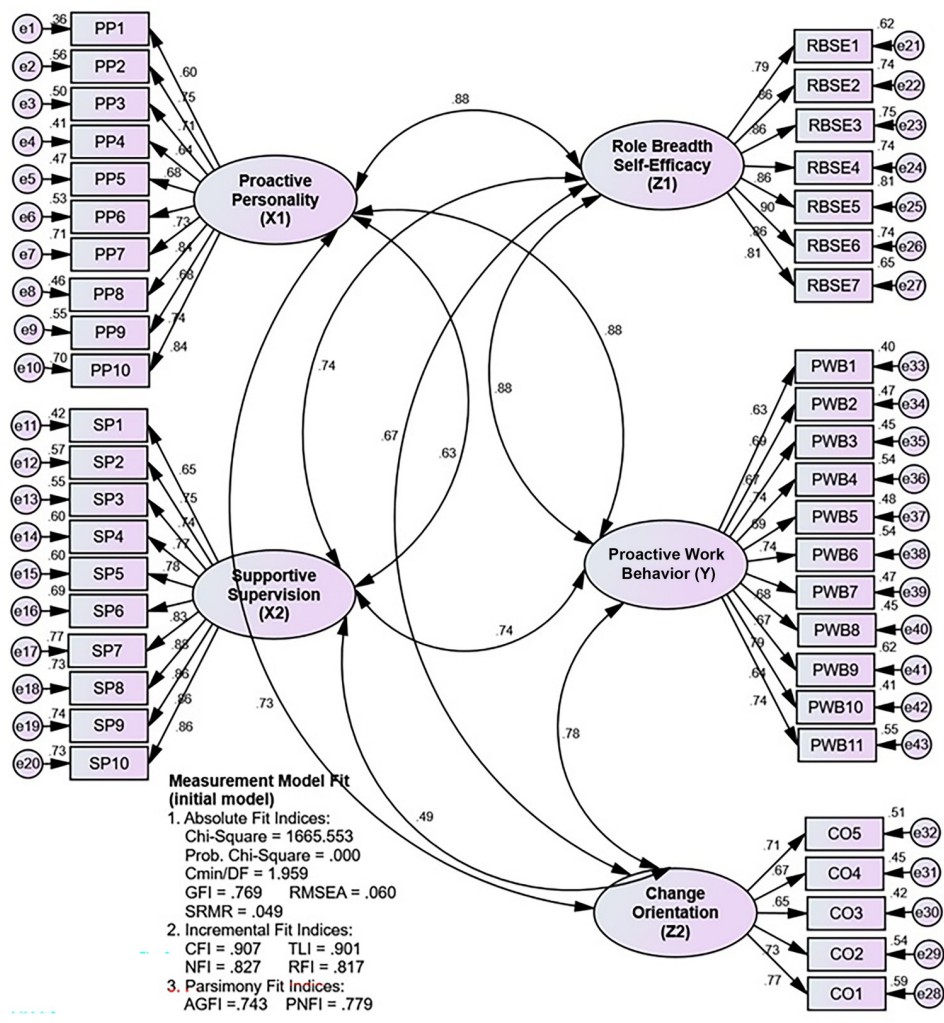

**Fig 2. The measurement model (initial model) is here.**

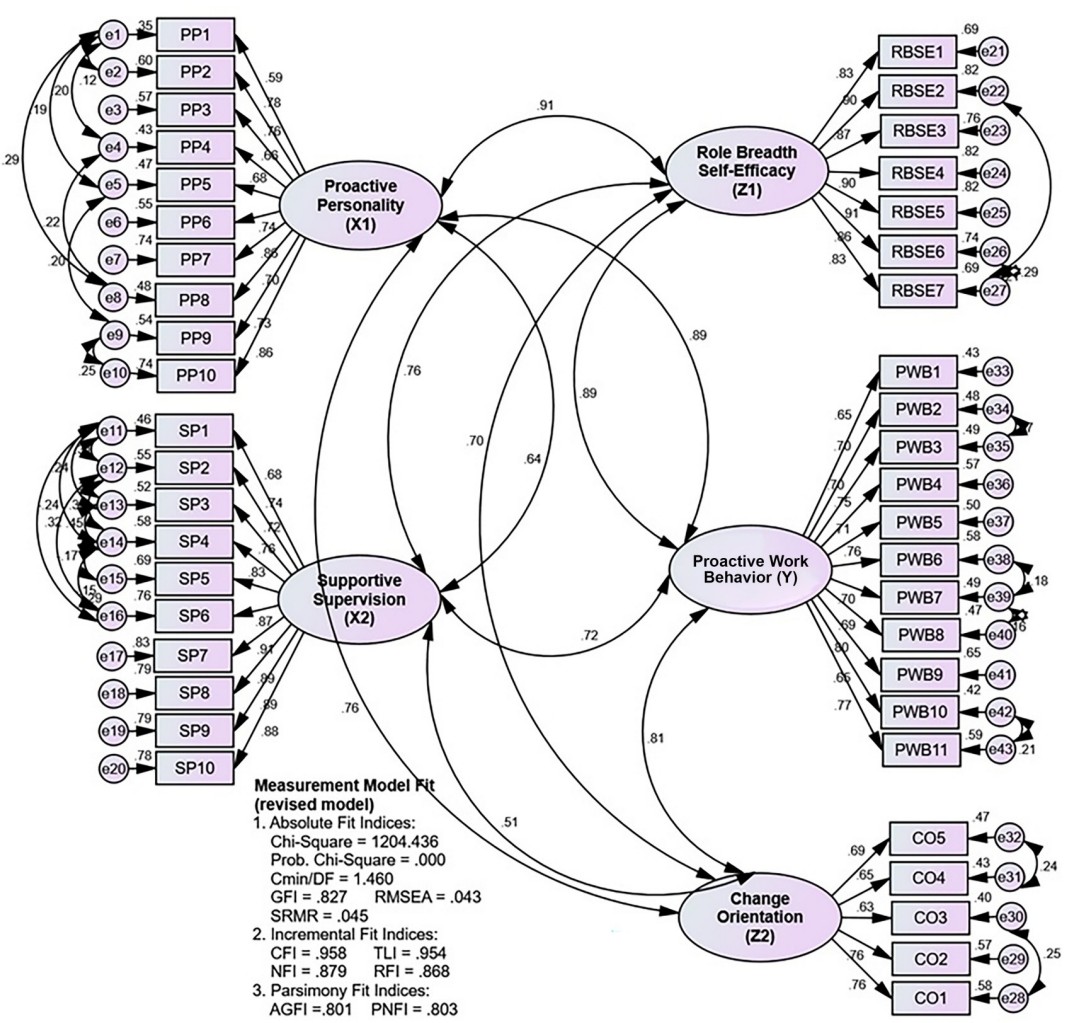

**Fig 3. The measurement model (revised model) is here.**

work behavior. Rehabilitation officers with a proactive personality exhibit initiative, creativity, and a readiness to adapt to challenging situations, such as those encountered in penitentiarys —specifically, dealing with issues like excess capacity, pandemic repercussions, and hurdles in rehabilitation implementation. The earlier-mentioned research problems, particularly concerning the deficiency of innovation and creativity in the rehabilitation process, directly correlate with the finding that a proactive personality is positively linked to proactive work behavior. This underscores the significance of enhancing the active personality of rehabilitation personnel to elevate the effectiveness of rehabilitation. Consequently, these findings offer a contribution to comprehending the dynamics of proactive work behavior within penitentiarys.

The resultant coefficient of influence is 0.169, which indicates that the more supportive the supervision, the more proactive the work behavior (H2 is accepted). Leader support (encouragement, accessibility, and non-interference) favorably influences employee work behavior [34]. The results of this study suggested that rehabilitation officers' supervisory support might provide a secure atmosphere for expressing creative and new ideas and give the necessary resources to do so successfully. When supportive supervision for innovation is high,

**Table 6. Construct validity test.**

| Variable | Indicator | Construct Validity | | |
|---|---|---|---|---|
| | | *Factor Loading* | AVE | Inf. |
| *Proactive Personality* | PP1 | 0,591 | 0,547 | Valid |
| | PP2 | 0,777 | | Valid |
| | PP3 | 0,757 | | Valid |
| | PP4 | 0,656 | | Valid |
| | PP5 | 0,683 | | Valid |
| | PP6 | 0,739 | | Valid |
| | PP7 | 0,862 | | Valid |
| | PP8 | 0,696 | | Valid |
| | PP9 | 0,733 | | Valid |
| | PP10 | 0,860 | | Valid |
| *Supportive Supervision* | SP1 | 0,678 | 0,675 | Valid |
| | SP2 | 0,745 | | Valid |
| | SP3 | 0,719 | | Valid |
| | SP4 | 0,762 | | Valid |
| | SP5 | 0,832 | | Valid |
| | SP6 | 0,871 | | Valid |
| | SP7 | 0,910 | | Valid |
| | SP8 | 0,888 | | Valid |
| | SP9 | 0,887 | | Valid |
| | SP10 | 0,883 | | Valid |
| *Role Breadth Self-Efficacy* | RBSE1 | 0,833 | 0,763 | Valid |
| | RBSE2 | 0,903 | | Valid |
| | RBSE3 | 0,870 | | Valid |
| | RBSE4 | 0,903 | | Valid |
| | RBSE5 | 0,907 | | Valid |
| | RBSE6 | 0,863 | | Valid |
| | RBSE7 | 0,833 | | Valid |
| *Change Orientation* | CO1 | 0,762 | 0,504 | Valid |
| | CO2 | 0,757 | | Valid |
| | CO3 | 0,682 | | Valid |
| | CO4 | 0,653 | | Valid |
| | CO5 | 0,689 | | Valid |
| *Proactive Work Behavior* | PWB1 | 0,653 | 0,516 | Valid |
| | PWB2 | 0,696 | | Valid |
| | PWB3 | 0,700 | | Valid |
| | PWB4 | 0,753 | | Valid |
| | PWB5 | 0,705 | | Valid |
| | PWB6 | 0,762 | | Valid |
| | PWB7 | 0,702 | | Valid |
| | PWB8 | 0,687 | | Valid |
| | PWB9 | 0,805 | | Valid |
| | PWB10 | 0,651 | | Valid |
| | PWB11 | 0,771 | | Valid |

rehabilitation officers are more likely to begin and maintain creative activity and coordinate their inventive efforts through proactive work behavior. Rehabilitation officers grappling with

**Table 7. Construct reliability test.**

| Variable | Construct Reliability (*Revised model*) | Information |
|---|---|---|
| *Proactive Personality* | 0,923 | Reliable |
| *Supportive Supervision* | 0,954 | Reliable |
| *Role Breadth Self-Efficacy* | 0,958 | Reliable |
| *Change Orientation* | 0,835 | Reliable |
| *Proactive Work Behavior* | 0,921 | Reliable |

challenges like overcapacity, pandemic impact, and hurdles in rehabilitation implementation are likely to exhibit increased initiative and creativity when adequately supported by their superiors. The identified research problems, particularly disruptions in rehabilitation activities due to the pandemic and the absence of innovation to surmount these obstacles, highlight the positive influence of supportive supervision on proactive work behavior. This implies that initiatives aimed at enhancing supervisory support for rehabilitation officers can prove effective in fostering initiative and creativity in their work performance. Consequently, these findings offer crucial insights for penitentiary managers, especially those overseeing Narcotics Penitentiary, to enhance supportive supervision practices.

The third hypothesis is accepted. Following earlier studies, the role breadth of self-efficacy has been validated as a key antecedent or predictor of proactive organizational work behavior [18,42]. Through role breadth of self-efficacy, the rehabilitation officers at the Narcotics Penitentiary demonstrated the ability to apply active thinking at work and proactively solve problems that were deemed important because they increased initiative and assumed responsibility, which is conducive to proactive work behavior. Overcoming research challenges, such as the deficiency in innovation and creativity in rehabilitation delivery amid issues like excess capacity, pandemic impact, and access barriers, can be achieved by reinforcing role breadth self-efficacy. The confidence to proactively engage in diverse roles can inspire the development of innovative and alternative solutions to support rehabilitation objectives. These findings suggest that penitentiary managers, particularly those overseeing Narcotics Prisons, should focus on boosting the confidence of rehabilitation officer by involving them in broader roles.

The resultant coefficient of effect is 0.313, which indicates that the greater the change orientation, the greater the proactive work behavior. Consequently, the fourth hypothesis is likewise valid (H4 accepted). According to a previous study, change orientation is positively associated with proactive work behavior [35]. The findings of this study indicated that rehabilitation officers could engage in several change orientation activities by bringing about changes, such as introducing new work methods and influencing organizational strategies, and changes in oneself, such as acquiring new skills to meet future demands, which leads to proactive work behavior. Rehabilitation officers possessing a change orientation are inclined to innovate in work practices and organizational strategies, fostering positive work behavior. Issues like the suspension of rehabilitation activities due to the pandemic, overcapacity, and obstacles in rehabilitation implementation can be addressed by reinforcing the change orientation of rehabilitation officers. Those with a robust change orientation are more proactive in devising innovative solutions to overcome hurdles and adapting their work methods to evolving circumstances. Consequently, these findings underscore the direct significance of promoting and supporting the development of a change orientation among rehabilitation officers in prison management, particularly in narcotics prisons.

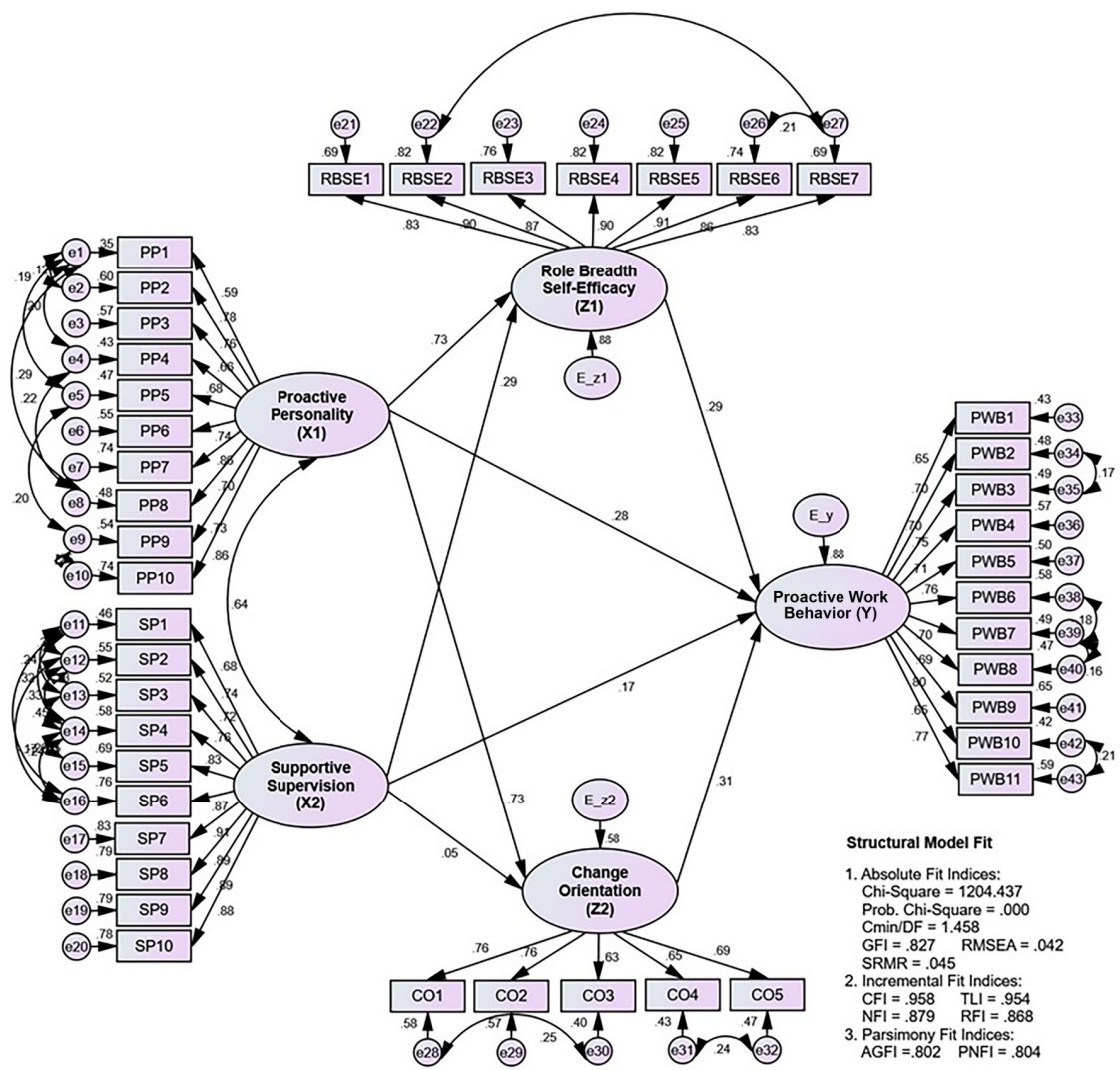

**Fig 4. Estimation results of structural equation modeling about here.**

**Table 8. Fit measure on structural model.**

| Fit Measure | | Structural Model | Critical Value | Decision |
|---|---|---|---|---|
| Absolute Fit Indices | Probability | 0,000 | ≤ 0,05 | Good fit |
| | Cmin/DF | 1,458 | ≤ 2,00 | Good fit |
| | GFI | 0,827 | ≥ 0,90 | Marginal fit |
| | RMSEA | 0,042 | ≤ 0,08 | Good fit |
| | SRMR | 0,045 | ≤ 0,05 | Good fit |
| Incremental Fit Indices | CFI | 0,958 | ≥ 0,95 | Good fit |
| | TLI | 0,954 | ≥ 0,95 | Good fit |
| | NFI | 0,879 | ≥ 0,90 | Marginal fit |
| | RFI | 0,868 | ≥ 0,90 | Marginal fit |
| Parsimony Fit Indices | AGFI | 0,802 | ≥ 0,90 | Marginal fit |
| | PNFI | 0,804 | ≥ 0,90 | Marginal fit |

**Table 9. Structural relationships between variables.**

| Hyp. | Structural Relations | | | Std. Estimate | C.R. | P value | Inf. |
|---|---|---|---|---|---|---|---|
| H$_1$ | Proactive Personality | → | Proactive Work Behavior | 0,279 | 2,281 | 0,023 | s |
| H$_2$ | Supportive Supervision | → | Proactive Work Behavior | 0,169 | 3,020 | 0,003 | s |
| H$_3$ | Role Breadth Self-Efficacy | → | Proactive Work Behavior | 0,286 | 2,338 | 0,019 | s |
| H$_4$ | Change Orientation | → | Proactive Work Behavior | 0,313 | 4,629 | 0,000 | s |
| H$_5$ | Proactive Personality | → | Role Breadth Self-Efficacy | 0,728 | 9,015 | 0,000 | s |
| H$_6$ | Supportive Supervision | → | Role Breadth Self-Efficacy | 0,290 | 6,396 | 0,000 | s |
| H$_7$ | Proactive Personality | → | Change Orientation | 0,728 | 7,112 | 0,000 | s |
| H$_8$ | Supportive Supervision | → | Change Orientation | 0,050 | 0,683 | 0,495 | ns |

Information: s (significant); ns (not significant).

The fifth hypothesis is valid (H5 accepted). It is consistent with previous studies, which indicated that a proactive personality is favorably associated with self-efficacy role breadth [39]. The present study also revealed that when rehabilitation officers speak to proactive personality, they refer to those who can take action, play a more flexible role, and make adjustments akin to mastery or control [3]. It might be the personal capacity of rehabilitation officers to promote role breadth self-efficacy by convincing individuals that they can perform a bigger role.

The sixth hypothesis is likewise acceptable (positive). According to the previous study, supportive supervision directly correlates with role breadth self-efficacy [3]. The results of this study indicate that supportive supervision is a major factor in the practical environment, as this support will be accompanied by a positive practice environment that encourages rehabilitation officers' professional autonomy in enhancing their professional competence and optimizing their performance, resulting in positive outcomes. Role breadth self-efficacy is advantageous for organizations.

The seventh hypothesis is valid (H7 accepted). Following the previous studies' findings, their research indicates that a proactive personality is closely associated with change orientation [3]. The findings of this study indicate that rehabilitation officers for narcotics addicts at the Narcotics Penitentiary with a change orientation in the organization provide positive job perceptions of individual initiatives that are also capable of encouraging adaptive behavior consistent with organizational goals and values that lead to proactive personality to improve

**Table 10. Indirect path structural relationships.**

| Hyp. | Indirect Path | Estimate | SE | C.R. | P-value | The nature of the mediator |
|---|---|---|---|---|---|---|
| H$_9$ | PP → RBSE → PWB | 0,281 | 0,125 | 2,243 | 0,025(s) | Partially mediation |
| H$_{10}$ | SP → RBSE → PWB | 0,076 | 0,035 | 2,167 | 0,031(s) | Partially mediation |
| H$_{11}$ | PP → CO → PWB | 0,308 | 0,080 | 3,839 | 0,000(s) | Partially mediation |
| H$_{12}$ | SP → CO → PWB | 0,014 | 0,022 | 0,658 | 0,511(ts) | No mediation |

PP: Proactive Personality.

SP: Supportive Supervision.

RBSE: Role Breadth Self-Efficacy.

CO: Change Orientation.

PWB: Proactive Work Behavior.

Information: s (significant); ts (not significant).

conditions. Inmates at the Narcotics Penitentiary are more inclined to alter their situations independently than to allow themselves to be molded by their surroundings.

The H8 was rejected. It does not align with the previous study, which states that supportive supervision is directly related to change orientation [47]. The results of this study indicated that narcotics addict rehabilitation officers at the Narcotics Prison without support from supportive supervision would remain focused on change orientation because narcotics addict rehabilitation officers at the Narcotics Penitentiary know that change orientation is very important in encouraging organizational change for a sustainable organization.

The ninth hypothesis is acceptable. The nature of the mediator is partially mediation, meaning that increasing proactive work behavior can only be done by increasing proactive personality. However, if it is also supported by strengthening the role breadth of self-efficacy, then proactive work behavior can increase even more. Employee development in the workplace fully mediates the relationship between proactive personality and creative behavior [48]. The results showed that role breadth self-efficacy is useful for rehabilitation officers for narcotics addicts at the Narcotics Penitentiary because they know that their belief in themselves is a state that can change with the organizational environment and employee experience, so they simultaneously need to be carried out together with a proactive personality to form proactive work behavior.

Likewise, the ninth hypothesis is acceptable (H10 accepted). In order to improve proactive work behavior, supportive supervision must be raised, but if role breadth self-efficacy is also strengthened, proactive work behavior may increase even further. According to a previous study, perceived organizational support and supportive supervision significantly moderate the relationship between teamwork behavior and affective commitment. It suggests that employees with more communication, cooperation, and collaboration (teamwork behavior) report higher affective commitment [48]. The results indicated that narcotics addict rehabilitation officers at the Narcotics Penitentiary with role breadth self-efficacy would be able to perform a certain set of tasks and feel motivated to do them, which, if conducted with narcotics addict rehabilitation officers at the Narcotics Penitentiary who received supportive supervision, would be more effective in promoting proactive work behavior.

The eleventh hypothesis is acceptable (H11 accepted). The nature of the mediator is known to be partially mediation, which means that increasing proactive work behavior can only be done by increasing proactive personality. However, if it is supported by strengthening change orientation, proactive work behavior can increase even more. Change orientation is related to proactive work behavior [35]. The results showed that the rehabilitation officers for narcotics addicts at the Narcotics Penitentiary with a change orientation would take a proactive personality towards problems and be involved in changes that create a tendency to be assertive and dominant. That way, rehabilitation officers with proactive personalities will be equipped with the right change process to create proactive work behavior.

The twelfth hypothesis cannot be accepted (H12 rejected). The results showed that the change in orientation of the narcotics addict rehabilitation officers at the Narcotics Prison was unable to support their supportive supervision in creating or improving proactive work behavior. It can occur because supportive supervision cannot influence strongly, helping them focus on change orientation, which is also important for proactive work behavior.

## Conclusions and implication

### Conclusion

This study's data analysis and discussion concluded that proactive personality significantly influences proactive work behavior, role breadth self-efficacy, and change orientation. In

addition, despite the mediation of role breadth self-efficacy and change orientation, a proactive personality considerably impacts proactive work behavior. Supportive supervision greatly benefits proactive work behavior and self-efficacy for role breadth but has little influence on change orientation. When mediated by role breadth self-efficacy, supportive supervision considerably impacts proactive work behavior. When mediated by change orientation, however, the effect is minimal. Change orientation and role breadth self-efficacy substantially influence proactive work behavior. Thus, under the influence of proactive personality, supportive supervision, role breadth self-efficacy, and change orientation, rehabilitation officers for narcotics addicts at the Narcotics Penitentiary can build and improve proactive work behavior. As a result, the rehabilitation officers at the Narcotics Penitentiary can carry out their duties to the fullest extent by using an idea process to develop ideas related to the COVID-19 epidemic. However, many penitentiarys have suspended rehabilitation operations due to the ban on outsiders. The study's findings significantly contribute to comprehending the factors influencing the proactive work behavior of rehabilitation officers. They underscore the importance of developing a proactive personality, enhancing supervisory support, and strengthening self-confidence in achieving rehabilitation goals and bolstering organizational efficiency.

## Theoretical implication

The findings underscore crucial theoretical connections between psychological variables and Proactive Work Behavior. Proactive personality strongly impacts proactive work behavior, role breadth self-efficacy, and change orientation, fostering innovation in the workplace. Developing proactive personality is vital for effective human resource management and achieving organizational goals. Supportive supervision is positively linked to improved proactive work behavior and higher role breadth self-efficacy, emphasizing the role of leadership in cultivating an environment conducive to innovation. Role breadth self-efficacy emerges as a key predictor of proactive work behavior, promoting initiative and responsibility. Strengthening this self-efficacy is a valuable strategy for enhancing proactivity and positive work outcomes. Change orientation positively influences proactive work behavior, highlighting the importance of fostering adaptability in human resource management strategies. These theoretical insights deepen our understanding of factors influencing proactive work behavior, providing a foundation for further theoretical development and valuable insights for human resource management research. Ultimately, officers, under the influence of proactive personality, supportive supervision, role breadth self-efficacy, and change orientation, can optimize their duties through innovative thinking and enhanced proactive work behavior.

## Managerial implications

The results can be used as recommendations by organization management regarding the effect of proactive personality and supportive supervision on proactive work behavior by mediating the role breadth of self-efficacy and changing the orientation of rehabilitation officers. It is known to suggest new ways of doing work and generate new ideas to increase initiative, encourage feedback and voice concerns among employees, give confidence in creating and developing ideas and solutions that can organize and implement actions to display skills in achieving goals, make changes related to policies, methodologies and work procedures that also occur in roles outside of authority, and create anticipatory actions that include preventing problems, seizing opportunities, and self-supporting efforts aimed at improving and changing the context or work situation and one's future. In addition, it is known that this study shows the lowest mean results on the dependent variable proactive work behavior, which states that officers have suggested ideas for improvement to managers, superiors, or others, with a score

of 4.04. Although this score is still high, it is the lowest among other indicators. So, rehabilitation officers are expected to take more initiative in suggesting important ideas to help improve work operations or the innovation process needed by managers, superiors, or others.

The research findings have important implications for the management of human resources in penitentiarys. Officers should actively engage in self-development by participating in training programs that specifically target the enhancement of initiative, creativity, and problem-solving abilities. It is vital to prioritize the provision of appropriate supervisory support, together with training for supervisors to provide constructive criticism. Personal development programs should bolster officers' self-assurance in taking on more expansive responsibilities. Management should cultivate an organizational culture that is favorable to change, by encouraging novel concepts and allocating room for innovation, supported by change management training.

By incorporating constructive supervisory techniques such as providing positive feedback and actively involving supervisors, an environment conducive to essential innovations can be fostered. Recognition and reward systems, which include promotions and incentives, are crucial for fostering favorable work behavior. Lastly, enhancing organizational communication pertaining to objectives, regulations, and modifications reduces ambiguity, promoting active engagement in organizational transformation. Through the use of these strategies, the management can foster a conducive work atmosphere, improve operational effectiveness, and successfully attain rehabilitation objectives.

## Limitations and recommendations for future research

This research has been carried out with maximum effort to the best of the researcher's knowledge and abilities. However, there are several limitations that need to be considered for future research, because this study itself has weaknesses that require improvement, such as the use of a cross-sectional research design. In order to provide more reliable results, it is recommended that future research use a longitudinal study design and apply time-lag analysis. Additionally, it should be noted that this study did not accommodate the role of moderators, which could potentially influence relationships. Therefore, further research is recommended to consider the role of these moderator variables. Future research is also expected to explore external influences more deeply to investigate the impact of external factors, such as changes in policy, community attitudes, or public opinion, on the proactive work behavior of working officers. Understanding how external forces shape proactive behavior can inform organizational strategies and interventions.

## Author Contributions

**Conceptualization:** A. Yuspahruddin, Hafid Abbas, Indra Pahala, Anis Eliyana.

**Data curation:** A. Yuspahruddin.

**Formal analysis:** Anis Eliyana.

**Funding acquisition:** Zaleha Yazid.

**Investigation:** Hafid Abbas.

**Methodology:** A. Yuspahruddin, Indra Pahala.

**Project administration:** A. Yuspahruddin, Indra Pahala.

**Resources:** Hafid Abbas, Indra Pahala.

**Software:** Zaleha Yazid.

**Supervision:** Hafid Abbas, Anis Eliyana, Zaleha Yazid.

**Validation:** Hafid Abbas, Indra Pahala, Anis Eliyana, Zaleha Yazid.

**Visualization:** Zaleha Yazid.

**Writing – original draft:** A. Yuspahruddin.

**Writing – review & editing:** A. Yuspahruddin.

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
