## [Decision Letter · Decision Letter 0]

14 Apr 2023

PONE-D-23-01402Fostering Proactive Work Behavior: Where to Start?PLOS ONE

Dear Dr. Eliyana,

Thank you for submitting your manuscript to PLOS ONE. After careful consideration, we feel that it has merit but does not fully meet PLOS ONE’s publication criteria as it currently stands. Therefore, we invite you to submit a revised version of the manuscript that addresses the points raised during the review process.

We look forward to receiving your revised manuscript.

Kind regards,

Arslan Ayub

Academic Editor

PLOS ONE

Journal Requirements:

4. Please ensure that you refer to Figure 1 in your text as, if accepted, production will need this reference to link the reader to the figure.

Reviewers' comments:

Reviewer's Responses to Questions

**Comments to the Author**

1. Is the manuscript technically sound, and do the data support the conclusions?

Reviewer #1: Partly

2. Has the statistical analysis been performed appropriately and rigorously? 

Reviewer #1: Yes

3. Have the authors made all data underlying the findings in their manuscript fully available?

Reviewer #1: No

4. Is the manuscript presented in an intelligible fashion and written in standard English?

Reviewer #1: Yes

5. Review Comments to the Author

Reviewer #1: The manuscript is interesting as they examined rehabilitation officers at the Narcotics Penitentiary in Sumatra, they presented very interesting fact, however the authors did not highlight the theoretical gap as the basis of developing their model. Please add basic theories as the reference for the research design.

Methodology: please justify further on the samplng technique used. This sentence " proportionate purposive sampling approach that involved obtaining samples based on specific criteria" needs further clarification.

Lack of conistency on the terms used, which one is correct among these three: proactive behavior, proactive personality and proactive attitude, as those terms are different.

6. PLOS authors have the option to publish the peer review history of their article (what does this mean?). If published, this will include your full peer review and any attached files.

Reviewer #1: No

---

## [Author Response · Author response to Decision Letter 0]

6 Jun 2023

Reviewer #1: 

The manuscript is interesting as they examined rehabilitation officers at the Narcotics Penitentiary in Sumatra, they presented very interesting fact, however the authors did not highlight the theoretical gap as the basis of developing their model. 

Please add basic theories as the reference for the research design.

Methodology: please justify further on the samplng technique used. This sentence " proportionate purposive sampling approach that involved obtaining samples based on specific criteria" needs further clarification.

Lack of conistency on the terms used, which one is correct among these three: proactive behavior, proactive personality and proactive attitude, as those terms are different.

Dear reviewer,

Thank you for your comment.

Basic theories have been added in the Literature Review section.

The term proportionate purposive sampling approach has been revised.

The use of terms has been standardized.

---

## [Decision Letter · Decision Letter 1]

5 Dec 2023

PONE-D-23-01402R1Fostering Proactive Work Behavior: Where to Start?PLOS ONE

Dear Dr. Eliyana,

Thank you for submitting your manuscript to PLOS ONE. After careful consideration, we feel that it has merit but does not fully meet PLOS ONE’s publication criteria as it currently stands. Therefore, we invite you to submit a revised version of the manuscript that addresses the points raised during the review process.

We look forward to receiving your revised manuscript.

Kind regards,

Bo Pu, Ph.D.

Academic Editor

PLOS ONE

Journal Requirements:

Additional Editor Comments (if provided):

this manuscript should be revised according the comments from the reviewers.

Reviewers' comments:

Reviewer's Responses to Questions

**Comments to the Author**

1. If the authors have adequately addressed your comments raised in a previous round of review and you feel that this manuscript is now acceptable for publication, you may indicate that here to bypass the “Comments to the Author” section, enter your conflict of interest statement in the “Confidential to Editor” section, and submit your "Accept" recommendation.

Reviewer #1: All comments have been addressed

Reviewer #2: (No Response)

2. Is the manuscript technically sound, and do the data support the conclusions?

Reviewer #1: Yes

Reviewer #2: No

3. Has the statistical analysis been performed appropriately and rigorously? 

Reviewer #1: Yes

Reviewer #2: Yes

4. Have the authors made all data underlying the findings in their manuscript fully available?

Reviewer #1: Yes

Reviewer #2: No

5. Is the manuscript presented in an intelligible fashion and written in standard English?

Reviewer #1: Yes

Reviewer #2: Yes

6. Review Comments to the Author

Reviewer #1: (No Response)

Reviewer #2: The abstract is important to represent the content of the article and it need to be improved. For example, the first part of the abstract should be improved providing a better introduction of the topic and the reason that you conduct this work.

In introduction section it is good to express the need for the study with the backlog of literature exists in the framework. What is the scientific contribution of your paper to the science? Can you please indicate scientific implications of your paper?

Literature review: Please review the previous studies. The literature review does not show a research gap.

How were the survey results verified?

The empirical section in many places, is devoid of interpretation and critical analysis of the research results. Most often, it is a description of the table's contents.

The results section should be separated from the discussion.

Discussion section: It would be appropriate to specify in more detail how this research differs from the already published paper that deals with a similar topic. To increase the significance of the results, the discussion part should embrace the differences and similarities among your findings and those of other scholars.

Conslucions: The section Conclusion can be expanded to include the limitations of this work and an indication of future research directions. Further applications of this proposed method should be discussed in detail, it may also good to add some real life cases, which can express the benefit of proposed method, further, and it may catch more attention.

7. PLOS authors have the option to publish the peer review history of their article (what does this mean?). If published, this will include your full peer review and any attached files.

Reviewer #1: **Yes: **Wiwiek Rabiatul Adawiyah

Reviewer #2: No

---

## [Author Response · Author response to Decision Letter 1]

16 Jan 2024

Reviewer #2: 

The abstract is important to represent the content of the article and it need to be improved. For example, the first part of the abstract should be improved providing a better introduction of the topic and the reason that you conduct this work.

Dear Reviewer,

Thank you for your comments.

Abstract has been improved as suggested

In introduction section it is good to express the need for the study with the backlog of literature exists in the framework. What is the scientific contribution of your paper to the science? Can you please indicate scientific implications of your paper?

Literature review: Please review the previous studies. The literature review does not show a research gap.

Dear Reviewer,

Thank you for your comments.

More statements have been added in the Introduction section. The research gap is also stated in the Introduction section.

How were the survey results verified?

The empirical section in many places, is devoid of interpretation and critical analysis of the research results. Most often, it is a description of the table's contents.

The results section should be separated from the discussion.

Discussion section: It would be appropriate to specify in more detail how this research differs from the already published paper that deals with a similar topic. To increase the significance of the results, the discussion part should embrace the differences and similarities among your findings and those of other scholars.

Dear Reviewer,

Thank you for your comments.

The Results section is intended to present the results of the analysis and findings. An in-depth interpretation of the results is presented in the Discussion section.

Conclusions: The section Conclusion can be expanded to include the limitations of this work and an indication of future research directions. Further applications of this proposed method should be discussed in detail, it may also good to add some real life cases, which can express the benefit of proposed method, further, and it may catch more attention.

Dear Reviewer,

Thank you for your comments.

The Conclusion section has been improved as suggested. The authors also decided to separate the Implication section into Theoretical Implication and Managerial Implication.

---

## [Decision Letter · Decision Letter 2]

2 Feb 2024

Fostering Proactive Work Behavior: Where to Start?

PONE-D-23-01402R2

Dear Dr. Eliyana,

We’re pleased to inform you that your manuscript has been judged scientifically suitable for publication and will be formally accepted for publication once it meets all outstanding technical requirements.

Kind regards,

Bo Pu, Ph.D.

Academic Editor

PLOS ONE

Additional Editor Comments (optional):

this manuscript should be published.

Reviewers' comments:

Reviewer's Responses to Questions

**Comments to the Author**

1. If the authors have adequately addressed your comments raised in a previous round of review and you feel that this manuscript is now acceptable for publication, you may indicate that here to bypass the “Comments to the Author” section, enter your conflict of interest statement in the “Confidential to Editor” section, and submit your "Accept" recommendation.

Reviewer #2: (No Response)

2. Is the manuscript technically sound, and do the data support the conclusions?

Reviewer #2: Partly

3. Has the statistical analysis been performed appropriately and rigorously? 

Reviewer #2: No

4. Have the authors made all data underlying the findings in their manuscript fully available?

Reviewer #2: No

5. Is the manuscript presented in an intelligible fashion and written in standard English?

Reviewer #2: Yes

6. Review Comments to the Author

Reviewer #2: (No Response)

7. PLOS authors have the option to publish the peer review history of their article (what does this mean?). If published, this will include your full peer review and any attached files.

Reviewer #2: No

---

## [Editor Report · Acceptance letter]

26 Mar 2024

PONE-D-23-01402R2 

PLOS ONE

Dear Dr. Eliyana, 

I'm pleased to inform you that your manuscript has been deemed suitable for publication in PLOS ONE. Congratulations! Your manuscript is now being handed over to our production team.

Kind regards, 

on behalf of

Dr. Bo Pu 

Academic Editor

PLOS ONE